# Native α-synuclein induces clustering of synaptic-vesicle mimics via binding to phospholipids and synaptobrevin-2/VAMP2

Jiajie Diao[1,2,3,4†], Jacqueline Burré[1†], Sandro Vivona[1,2,3,4], Daniel J Cipriano[1,2,3,4], Manu Sharma[1], Minjoung Kyoung[1,2,3,4‡], Thomas C Südhof[1,4*], Axel T Brunger[1,2,3,4*]

[1]Department of Molecular and Cellular Physiology, Stanford University, Stanford, United States; [2]Department of Structural Biology, Stanford University, Stanford, United States; [3]Departments of Photon Sciences, and Neurology and Neurological Sciences, Stanford University, Stanford, United States; [4]Howard Hughes Medical Institute, Stanford University, Stanford, United States

**Abstract** α-Synuclein is a presynaptic protein that is implicated in Parkinson's and other neurodegenerative diseases. Physiologically, native α-synuclein promotes presynaptic SNARE-complex assembly, but its molecular mechanism of action remains unknown. Here, we found that native α-synuclein promotes clustering of synaptic-vesicle mimics, using a single-vesicle optical microscopy system. This vesicle-clustering activity was observed for both recombinant and native α-synuclein purified from mouse brain. Clustering was dependent on specific interactions of native α-synuclein with both synaptobrevin-2/VAMP2 and anionic lipids. Out of the three familial Parkinson's disease-related point mutants of α-synuclein, only the lipid-binding deficient mutation A30P disrupted clustering, hinting at a possible loss of function phenotype for this mutant. α-Synuclein had little effect on $Ca^{2+}$-triggered fusion in our reconstituted single-vesicle system, consistent with in vivo data. α-Synuclein may therefore lead to accumulation of synaptic vesicles at the active zone, providing a 'buffer' of synaptic vesicles, without affecting neurotransmitter release itself.

**\*For correspondence:** brunger@stanford.edu (ATB); tcs1@stanford.edu (TCS)

†These authors contributed equally to this work

‡**Present address:** Department of Chemistry and Biochemistry, University of Maryland, Baltimore County, Baltimore, United States

## Introduction

α-Synuclein (α-Syn) is an abundant presynaptic protein that is expressed throughout the central nervous system (CNS) and associated with synaptic vesicles (*Maroteaux et al., 1988*; *Jensen et al., 1998*). Pathologically, missense mutations (A30P, E46K, A53T) of α-Syn (*Polymeropoulos et al., 1997*; *Kruger et al., 1998*; *Zarranz et al., 2004*) and duplications and triplications of the gene that encodes α-Syn (*Singleton et al., 2003*; *Chartier-Harlin et al., 2004*) are linked to early onset Parkinson's disease (PD). Moreover, Lewy bodies observed in Parkinson's disease, dementia with Lewy bodies, and other neurodegenerative diseases contain high concentrations of α-Syn aggregates (*Spillantini et al., 1997*; *Hardy and Gwinn-Hardy, 1998*; *Spillantini et al., 1998*). In aqueous solution, α-Syn is a largely unfolded protein and has a propensity to aggregate in a nucleation-dependent manner, forming β-sheet rich amyloid-like fibrils (*Serpell et al., 2000*; *Zhao et al., 2011*; *Burré et al., In press*). Fibrils, protofibrils and/or large oligomers are believed to be cytotoxic (*Conway et al., 2000*; *Goldberg and Lansbury, 2000*; *Bucciantini et al., 2002*), and may be responsible, at least in part, for the neurodegeneration observed in diseases featuring Lewy bodies. Physiologically, α-Syn increases the rate of SNARE-complex formation at the synapse via binding to the SNARE protein synaptobrevin-2 (also called VAMP2, vesicle associated membrane protein 2) and to phospholipids (*Burré et al., 2010*).

In adult canaries and zebra finches, α-Syn expression correlates with plasticity in the developing song control system (*Clayton and George, 1998*). However, deletion of α-Syn in mice has been

**eLife digest** The central nervous system coordinates many different activities by sending instructions to large numbers of cells and, simultaneously, processing all the signals that are sent back to the brain. All these messages are carried by electrical pulses that travel along chains of neurons, with neurotransmitter molecules enclosed inside synaptic vesicles conveying the messages across the synapses between neurons. A protein called α-synuclein is thought to have a role in the transport of neurotransmitter molecules across synapses, but the details of its involvement are not fully understood.

Mutations in the gene that codes for α-synuclein, and also duplications and triplications of this gene, are known to lead to an increased risk of early onset Parkinson's disease, a condition where the central nervous system degenerates. Moreover, the Lewy bodies found in the neurons of patients with Parkinson's disease contain high concentrations of α-synuclein. Again, however, none of this is fully understood.

Diao et al. have shed new light on these questions by creating synthetic vesicles to mimic what happens in real synapses, and using optical microscopy to observe the behaviour of these vesicles. They found that native α-synuclein (and another set of membrane proteins) increases the availability of synthetic vesicles at the synapse by causing them to cluster together. In a second experiment, Diao et al. showed that native α-synuclein does not decrease calcium-triggered fusion between membranes, the process that releases neurotransmitter into the synaptic cleft. In contrast, it is known that pathogenic α-synuclein aggregates directly interfere with the release of the neurotransmitter molecules. Moreover, when Diao et al. used a particular mutant form of α-synuclein that is associated with Parkinson's disease, the vesicles did not form clusters. If these results are confirmed in vivo, the role played by native α-synuclein in the central nervous system, and the connection between α-synuclein and Parkinson's disease, will be much clearer.

reported to show either no or very small and opposing effects on neurotransmitter release (*Abeliovich et al., 2000*; *Cabin et al., 2002*; *Chandra et al., 2004*; *Liu et al., 2004*; *Yavich et al., 2004*; *Unger et al., 2006*; *Senior et al., 2008*; *Burré et al., 2010*; *Garcia-Reitbock et al., 2010*; *Scott et al., 2010*; *Anwar et al., 2011*). Similarly, overexpression of α-Syn either in transgenic mice or using stereotactic injections of adeno-associated virus have resulted in conflicting effects on neurotransmitter release, although chronic overexpression nearly always leads to neurodegeneration (*Liu et al., 2004*; *Larsen et al., 2006*; *Burré et al., 2010*; *Nemani et al., 2010*; *Gaugler et al., 2012*).

Structurally, α-Syn is a 14-kDa protein composed of an amphipathic, positively charged 100 residue N-terminal domain with a lysine-rich N-terminus that binds reversibly to anionic membranes (*Rhoades et al., 2006*), and a 40-residue highly acidic C-terminal domain that interacts with the N-terminal sequence of synaptobrevin-2 (*Burré et al., 2010*). Recombinant, erythrocyte, and brain α-Syn is predominantly monomeric in solution with a smaller fraction of multimeric species, and is largely unstructured with a small (approximately 21–24%) helical contribution (*Fauvet et al., 2012*; *Burré et al., 2013*). At high concentrations (~500 μM) and in the presence of β-octylglucoside, nuclear magnetic resonance spectra showed weak intramolecular NOEs in regions of α-helical propensity but no tertiary interactions, reminiscent of a molten globule (*Wang et al., 2011*). The interaction with anionic lipids induces α-helicity in α-Syn, as revealed by CD spectroscopy (*Davidson et al., 1998*; *Jo et al., 2000*). Moreover, solution NMR studies, EPR analyses, calorimetry, and single molecule fluorescence resonance transfer experiments in the presence of SDS micelles and small unilamellar vesicles revealed that the N-terminal 100 residues form a broken amphipathic α-helix upon binding to anionic lipids (*Bussell and Eliezer, 2003*; *Chandra et al., 2003*; *Ulmer et al., 2005*; *Jao et al., 2008*; *Ferreon et al., 2009*; *Lokappa and Ulmer, 2011*; *Maltsev et al., 2013*). Binding of α-Syn to the membrane is transient, and the C-terminal 40 residues of α-Syn are unstructured and not bound to the membrane (*Bodner et al., 2009*). N-terminal acetylation of α-Syn does not induce significant structure in recombinant α-Syn in solution (i.e., absence of detergents or lipids), but further enhances its binding to anionic membranes and induced α-helicity (*Maltsev et al., 2012*). Although initially largely monomeric, purified brain α-Syn associates into partially folded multimers and aggregates in a time-dependent manner (*Burré et al., 2013*). It is likely that these different monomeric and oligomeric states exist in

an equilibrium in vivo, which is modulated by a variety of factors and interactions, such as membrane binding. This labile mixture of conformations provides a potential explanation for why α-Syn is so susceptible to pathological aggregation and fibril formation as observed in multiple neurodegenerative disorders (*Devine et al., 2011*; *Martin et al., 2011*).

To investigate the effect of native α-Syn on synaptic vesicle fusion, we employed a recently developed in vitro system (*Kyoung et al., 2011*; *Diao et al., 2012a*) that allowed us to measure the effect of α-Syn on $Ca^{2+}$-triggered vesicle fusion with reconstituted neuronal SNAREs, synaptotagmin-1, and complexin-1 on a sub-sec timescale. We found that native α-Syn does not significantly affect the efficiency and kinetics of $Ca^{2+}$-triggered synaptic protein-mediated fusion itself. Rather, α-Syn induces clustering of vesicles with reconstituted synaptobrevin-2 or with both reconstituted synaptobrevin-2 and synaptotagmin-1. The clustering effect is dependent on specific binding to both anionic lipid membranes and synaptobrevin-2 as revealed by deletion and mutagenesis studies, including the Parkinson's disease mutant A30P that disrupts lipid binding. Furthermore, clustering is induced by both recombinant and brain-purified native α-Syn, both of which are initially in a largely monomeric state in the absence of membranes. This clustering or lack thereof may account for the increase in SNARE-complex levels observed upon overexpression of native α-Syn (*Burré et al., 2010*), and, conversely, the decrease in SNARE-complex assembly in αβγ-Syn triple knockout mice (*Burré et al., 2010*).

---

**Box 1.** Definitions

**v-vesicle**: proteoliposome with reconstituted synaptobrevin-2 in all experiments except *Figures 1, 2, and 4D* where both synaptobrevin-2 and synaptotagmin-1 were reconstituted. We also refer to these vesicles as "synaptic-vesicle mimics".

**t-vesicle**: proteoliposome with reconstituted syntaxin-1A and SNAP-25A. We also refer to these vesicles as "plasma-membrane mimics".

**v-/t-vesicle association**: fluorescent spot arising from labeled v-vesicles (or small clusters of v-vesicles) that bind to a surface with immobilized t-vesicles, using the protocol described in the 'Single vesicle-vesicle $Ca^{2+}$-triggered fusion experiments' section in 'Materials and methods'.

**v-/v-vesicle cluster**: fluorescent spot arising from labeled v-vesicles that bind to a surface with immobilized v-vesicles, using the protocol described in the 'Single-vesicle clustering experiments' section in 'Materials and methods'.

---

## Results

### Native α-Syn has little effect on $Ca^{2+}$-triggered synaptic-protein mediated v-/t-vesicle fusion

To address the question of whether native α-Syn influences $Ca^{2+}$-triggered synaptic-protein mediated membrane fusion, we employed our recently developed in vitro single vesicle–vesicle fusion system using vesicles with reconstituted synaptic proteins (*Kyoung et al., 2011*; *Diao et al., 2012a*; *Kyoung et al., 2012*). Free-floating 'v-vesicles' (with reconstituted synaptobrevin-2 and synaptotagmin-1) mimicked synaptic vesicles, while surface-immobilized 't-vesicles' (with reconstituted syntaxin-1 and SNAP-25) mimicked the presynaptic plasma membrane (*Figure 1A*). A defined volume of v-vesicle solution was incubated with t-vesicles together with complexin-1, establishing a metastable state at zero $Ca^{2+}$. We tested the effect of recombinant α-Syn with our system by adding it to the v-vesicle incubation stage. For the v-vesicles that bound, native α-Syn had little effect on the efficiency (defined as the number of fusion events per docked v-vesicles) and the kinetics of $Ca^{2+}$-triggered fusion upon injection of 500 μM $Ca^{2+}$ (*Figure 1B*), and it did not alter the corresponding cumulative fusion distribution plot (*Figure 1C*).

### α-Syn induces v-vesicle clustering

Although native α-Syn showed no effect on $Ca^{2+}$-triggered fusion activity, we noticed that α-Syn reduced the number of fluorescent spots that appeared when v-vesicles bound to immobilized t-vesicles during the incubation stage at zero $Ca^{2+}$ (*Figure 2A*). However, the addition of native α-Syn produced the emergence of brighter spots between 1.5 and 6 a.u. in the fluorescence emission histogram (insert, *Figure 2B*), along with occasional very large fluorescent spots (black arrow, lower right image in *Figure 2A*). The major peak

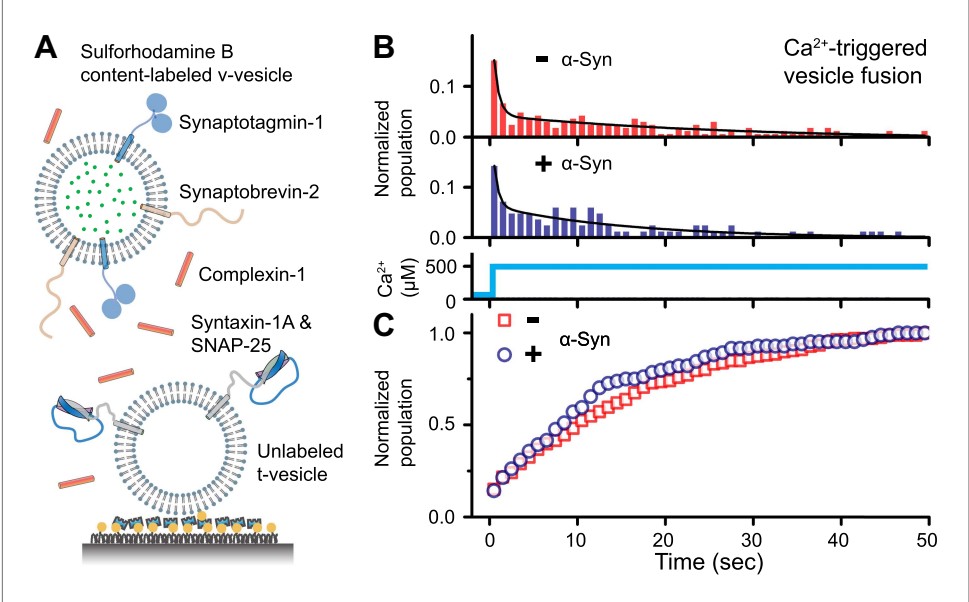

**Figure 1**. Native α-Syn has no effect on $Ca^{2+}$-triggered v-/t-vesicle fusion. (**A**) Experimental scheme of our single v-/t-vesicle content mixing system (***Kyoung et al., 2011***; ***Kyoung et al., 2012***), with improvements described in (***Diao et al., 2012a***), and further modifications described in 'Materials and methods'. (**B**) and (**C**) α-Syn has little effect on the probability of triggered vesicle fusion upon injection of 500 μM $Ca^{2+}$. Panels show histograms of the occurrence (**B**) and the corresponding cumulative distribution (**C**) of $Ca^{2+}$-triggered complete fusion (content mixing) in presence or absence of 2 μM α-Syn. In the absence of α-Syn, we observed 166 fusion events out of ~2000 docked v-vesicles (identified as fluorescent spots that were present prior to $Ca^{2+}$ injection), whereas in the presence of α-Syn, we observed 84 fusion events out of ~1300 docked v-vesicles within the observation period of 50 s. The time-binning was 1 s. Black lines are fits to bi-exponential decay functions over the entire observation period of 50 s. In the absence of α-Syn, the fitted function is $f(t) = -0.0046 + 0.048\,e^{-t/27.1} + 0.26\,e^{-t/0.58}$, whereas in the presence of α-Syn, the fitted function is $f(t) = -0.001 + 0.064\,e^{-t/15.4} + 0.22\,e^{-t/0.51}$ with t in sec.

(<1.5 a.u.) at low fluorescence intensity corresponds to single donor v-vesicles docked to the imaging surface, while the high intensity tail (>2 a.u.) corresponds to small clusters of v-vesicles docked to the surface. In the presence of α-Syn, we thus observed a reduction of the single vesicle population and a concomitant increase of a population of vesicle clusters. We note that the same finite volume and concentration of individual v-vesicles was used during the incubation stage with and without native α-Syn. Thus, the observed decrease in the total number of fluorescent spots (referred to as 'v-/t-vesicle associations') in the presence of native α-Syn (***Figure 2A***) can be explained by a reduction of the number of free-floating particles by clustering of individual v-vesicles in the finite volume of the sample chamber (***Figure 2C***).

We next confirmed the clustering activity of native α-Syn on v-vesicles with a different assay that was designed to directly probe the interaction between v-vesicles (***Figure 3A***). One population of v-vesicles was anchored to an imaging surface, while the other population was free-floating. Both vesicle populations were labeled with spectrally distinct fluorescent lipid analogues allowing tracking of the location of the free-floating vesicles and assessment of the surface coverage of the immobilized vesicles, respectively. Interactions between both populations of v-vesicles were measured by counting the number of initially free-floating v-vesicles that bound to the immobilized v-vesicles. We refer to this quantity as 'clustering' between vesicles of the same type (v-/v-vesicle clustering). The protocol of this clustering experiment included an incubation step with native α-Syn followed by buffer exchange, in order to measure the interaction between immobilized v-vesicles with bound α-Syn and free-floating v-vesicles without α-Syn (***Figure 3B***). This procedure prevented formation of the large fluorescent spots that we observed in the v-/t-vesicle fusion experiments, and it also reduced the likelihood of non-specific vesicle clustering by α-Syn oligomerization.

Using this clustering assay, we confirmed that native α-Syn increases the clustering of v-vesicles in a concentration-dependent manner (***Figure 4A***). We note that the specified α-Syn concentration refers

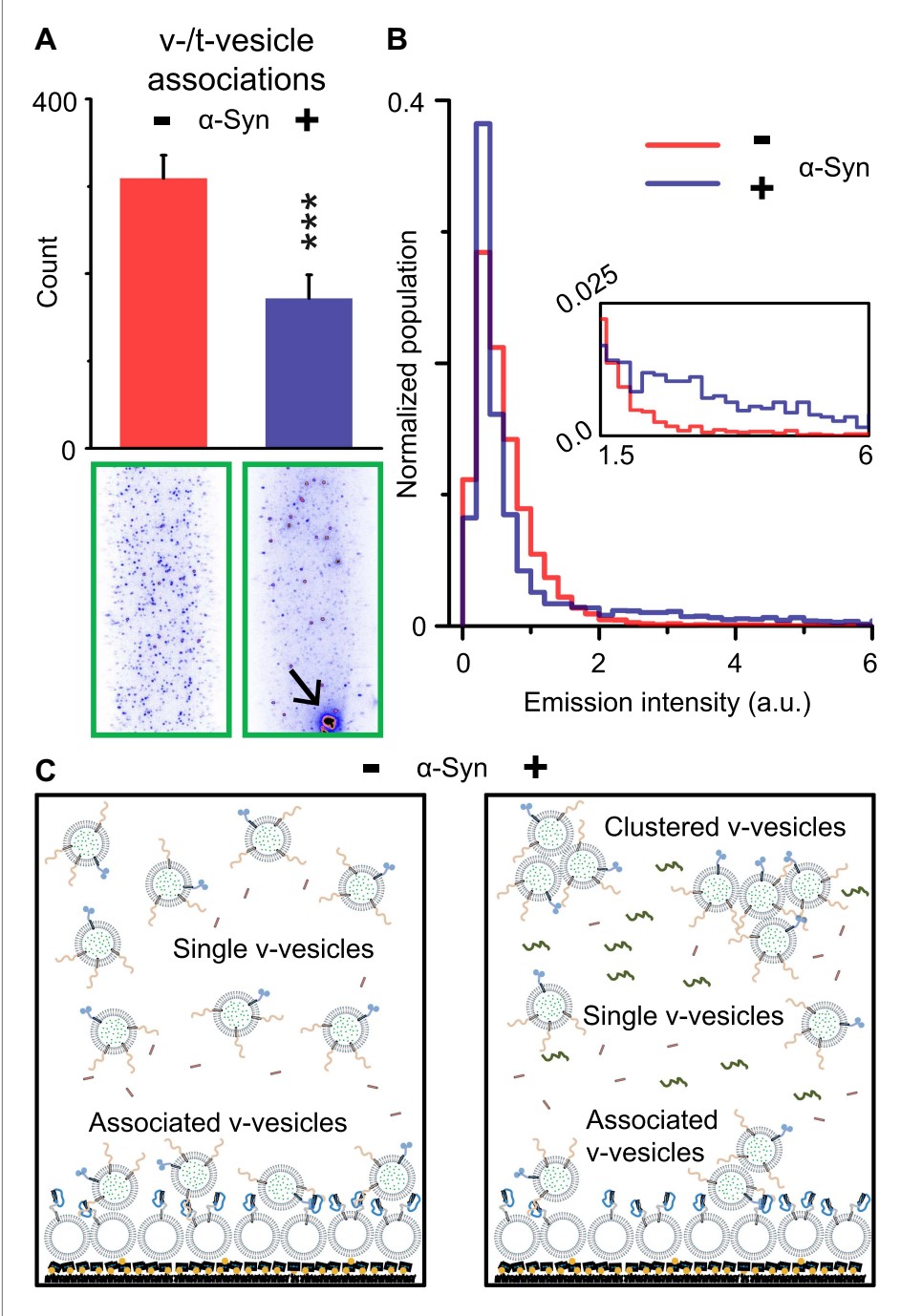

**Figure 2**. Native α-Syn decreases the number of v-/t-vesicle associations. (**A**) Plotted is the number of content dye fluorescent spots in presence of 2 µM complexin-1 and with or without 2 µM α-Syn (tag-free construct); example images are shown in the lower two panels. Scoring ('counts') excluded the few very large fluorescent spots that appeared in the presence of α-Syn; an example is shown in the lower right panel, marked by a black arrow. Error bars are standard deviations from 10 random imaging locations within the same sample channel. *** indicates p<0.001 by the Student's t-test. (**B**) Distribution of the fluorescence intensity of all fluorescent spots as shown in panel (**A**). In the presence of α-Syn, the number of low intensity fluorescent spots (up to 1.5 a.u.) decreased, which was accompanied by an increase in higher intensity fluorescent spots (see inset) due to the formation of v-vesicle clusters. The very large fluorescent spot (black arrow in the lower right image in panel (**A**)) was excluded in this distribution. (**C**) Illustration that clustering of v-vesicles induced by α-Syn reduces the effective number of fluorescent spots (corresponding to docked single or multiples of v-vesicles) in a finite volume since the starting concentration of individual v-vesicles was identical with and without α-Syn.

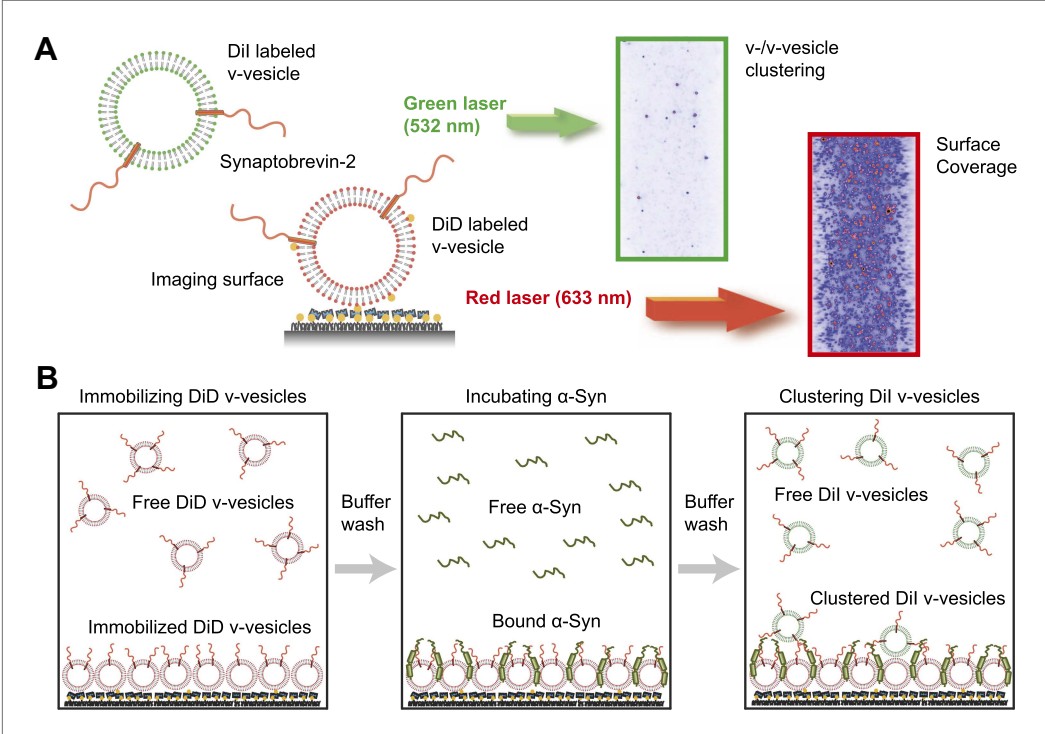

**Figure 3**. Native α-Syn induces clustering of synaptobrevin-2 v-vesicles. (**A**) Experimental scheme of our single-vesicle assay for monitoring clustering of synaptobrevin-2 v-vesicles. A saturated layer of DiD-labeled v-vesicles with reconstituted synaptobrevin-2 was immobilized on an imaging surface via biotin/neutravidin interactions. Free-floating DiI-labeled vesicles with reconstituted synaptobrevin-2 were injected into the sample chamber ('Materials and methods'). The number of v-/v-vesicle interactions (clustering) in presence or absence of α-Syn was determined by counting the number of spots arising from fluorescence emission of DiI upon excitation at 532 nm. (**B**) Experimental flow: 1. Immobilization of DiD-labeled v-vesicles on the imaging surface through biotin/neutravidin interactions. 2. Buffer exchange. 3. α-Syn incubation of the surface with immobilized DiD v-vesicles. 4. Buffer exchange, removing unbound or weakly bound α-Syn molecules. 5. Injection of DiI-labeled v-vesicles without α-Syn. Following another buffer exchange, the channels were imaged on the microscope.

to the incubation stage with immobilized v-vesicles (*Figure 3B*, middle panel). Subsequent to the incubation stage, a buffer exchange with α-Syn-free buffer was performed for 6 s which removed unbound α-Syn. Moreover, it is likely that the buffer exchange also removed some of the v-vesicle-bound α-Syn molecules since kinetic studies by NMR spectroscopy revealed that α-Syn binding to membranes is transient with exchanges occurring in the range of 1–10 s$^{-1}$ (*Bodner et al., 2009*). Thus, the effective α-Syn concentration during the final clustering step (*Figure 3B*, right panel) is lower than 2–20 µM, as only bound α-Syn is present at this stage. Indeed, as *Figure 4A* shows, v-vesicle clustering has not reached saturation at 2 µM α-Syn incubation concentration, so v-vesicles were not fully covered with α-Syn molecules at that particular α-Syn incubation concentration.

As another independent confirmation of v-vesicle clustering, a cryo-electron microscopy experiment revealed large v-vesicle clusters in the presence of native α-Syn (*Figure 5*). Note, that the vesicle concentration used for the cryo-EM experiment was 100-fold higher and the effective α-Syn concentration was also higher than that used for the single-vesicle clustering experiments, explaining the formation of the fairly large clusters in the cryo-EM experiment at that concentration. V-vesicles clustering does not occur in the absence of α-Syn (*Kyoung et al., 2011*) and is therefore specific to native α-Syn. It is also entirely dependent on the presence of synaptobrevin-2, as lack of synaptobrevin-2 (*Figure 4A*), or truncation of C-terminal residues of α-Syn that bind to synaptobrevin-2 (*Burré et al., 2010*) but do not participate in lipid binding of α-Syn (*Burré et al., 2010, 2012*) significantly reduced clustering of v-vesicles (*Figure 4B,C*). Furthermore, clustering of v-vesicles depended on the presence of negatively charged (anionic) lipids (*Figure 4A*, right most condition), which is known to induce

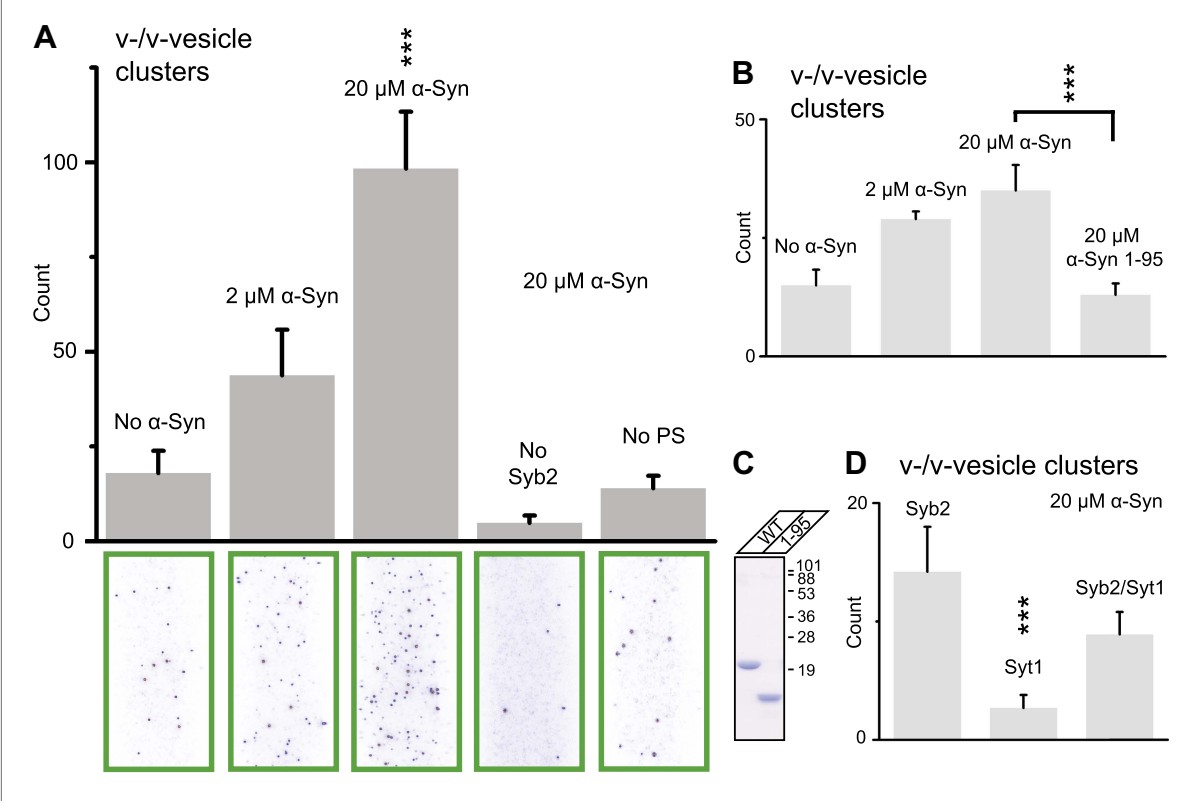

**Figure 4**. Native α-Syn (tag-free construct) promotes v-vesicle clustering by binding to both synaptobrevin-2 and anionic membranes. (**A**) α-Syn increases the number of interacting DiI-labeled v-vesicles on the imaging surface in a concentration-dependent manner. Bar graph: quantitation of interacting vesicles. Bottom panel: representative fluorescence images of interacting vesicles on the imaging surface. Error bars are standard deviations from 15 random imaging locations in the sample channel obtained. (**B**) Number of interactions between free-floating and immobilized v-vesicles in presence of wildtype α-Syn and its mutant that does not bind to synaptobrevin-2 (α-Syn[1-95] with myc-tag) at indicated concentrations. (**C**) Purified α-Syn wildtype and α-Syn[1-95] (with myc-tag) expressed in bacteria were analyzed by SDS-PAGE and Coomassie staining. (**D**) Number of interactions in the presence of 20 μM α-Syn (myc-tag construct). At variance to panels (**A**, **B**, and **C**), both free-floating and immobilized v-vesicles were reconstituted with synaptobrevin-2 (Syb2), synaptotagmin-1 (Syt1), or both (Syb2/Syt1). Note, that the slight variation in the 'absolute' number of observed fluorescent spots between comparable experiments in panels (**A**, **B**, and **D**) arises mainly from tagged vs untagged versions of α-Syn, as well as from different liposome and imaging surface preparations. Yet, the 'relative' differences were statistically similar for different protein preparations. In all panels, error bars are standard deviations from 10 random imaging locations in the same sample channel, and *** indicates p<0.001 by the Student's t-test.

α-helix formation in native α-Syn and lipid binding of α-Syn. As a control, and in agreement with previous studies (*Davidson et al., 1998*; *Jo et al., 2000*; *Eliezer et al., 2001*), we observed α-helicity in the CD spectrum of native α-Syn in the presence of protein-free vesicles with the same lipid composition used for our fusion and v-vesicle clustering experiments (*Figure 6*). Taken together, our clustering experiments suggest that native α-Syn has to bind to both, synaptobrevin-2 and anionic lipids, in order to cluster v-vesicles, confirming the protein- and anionic-lipid specificity of this phenomenon.

In the next experiment, we asked if the inclusion of synaptotagmin-1 in the synaptobrevin-2 vesicles would alter v-vesicle clustering induced by native α-Syn (*Figure 4D*). We found that v-vesicle clustering induced by native α-Syn was similar with and without synaptotagmin-1 within experimental error. Furthermore, v-vesicles with only reconstituted synaptotagmin-1 did not cluster, consistent with the requirement of synaptobrevin-2 for native α-Syn-induced v-vesicle clustering (*Figure 4A*). For the remaining experiments we therefore used v-vesicles that only contained synaptobrevin-2.

### V-vesicle clustering ability of native α-Syn is reduced by the Parkinson's disease mutant A30P

To further investigate the membrane-binding role of native α-Syn on v-vesicle clustering, and to extend our findings to PD, we performed experiments with the three PD-related mutants A30P, E46K, and

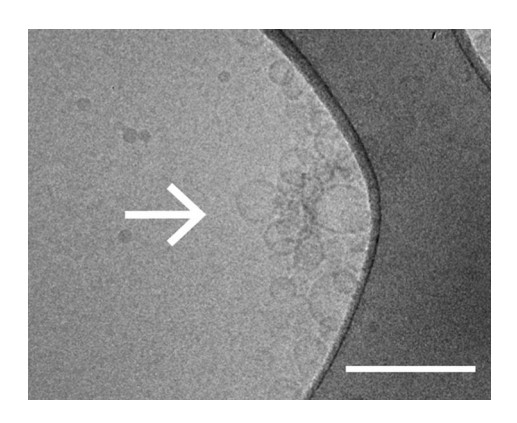

**Figure 5**. Cryo-electron microscopy image of clusters of synaptobrevin-2 v-vesicles as induced by native α-Syn. Scale bar is 200 nm. The dark feature is the holey carbon grid. In contrast, reconstituted v-vesicles in the absence of α-Syn do not form clusters (**Kyoung et al., 2011**).

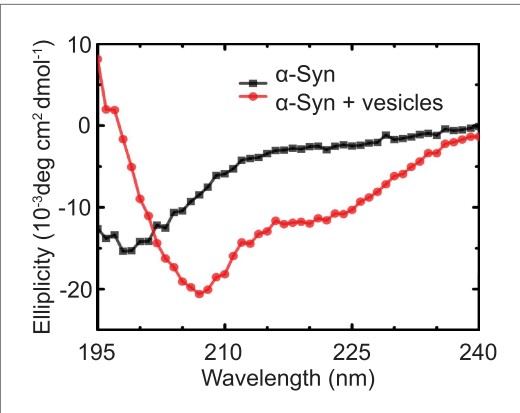

**Figure 6**. Native α-Syn undergoes a conformational transition from a predominantly unstructured state in solution to an α-helical state upon binding to protein-free vesicles as measured by CD spectroscopy. The protein-free vesicles had the same lipid composition as the v-vesicles used throughout this work. The molar protein-to-lipid ratio was 1:530, and tag-free wildtype α-Syn was used.

A53T (**Figure 7A**). While all these mutants retain the ability to bind synaptobrevin-2, only A30P shows reduced membrane binding (**Burré et al., 2012**). We measured the effect of these mutations on membrane-binding using a lipid flotation assay (**Figure 7B**), and on v-vesicle clustering using single vesicle microscopy (**Figure 7C**). Out of the mutants tested, only A30P showed ~ 50% decrease in membrane binding and v-vesicle clustering compared to native wildtype α-Syn. Thus, the effect of the three mutations in v-vesicle clustering correlated well with their effect on native α-Syn binding to PS-containing vesicles. This suggests that membrane-binding by native α-Syn is essential for its clustering activity.

To confirm that v-vesicle clustering induced by native α-Syn depends on lipid binding, we also measured the effect of the PD mutants on the number of v-/t-vesicle associations (**Figure 7D**). Only the A30P mutant increased the number of v-/t-vesicle associations. As mentioned above, the reduction of the number of v-/t-vesicle associations by native wildtype α-Syn is a consequence of the finite volume used during the incubation stage and the reduction of the number of free particles in the sample chamber by v-vesicle clustering. Thus, the numbers of v-vesicle clusters (**Figure 7C**) and v-/t-vesicle associations (**Figure 7D**) are anti-correlated.

## Native brain-purified α-Syn clusters v-vesicles

In order to test whether native brain-purified α-Syn has the same activity as native recombinant α-Syn in clustering v-vesicles, we purified native α-Syn from mouse brain without detergents or denaturants using multiple chromatography steps to a purity of >90% (**Burré et al., In press**). We found that the induction of v-vesicle clustering by native brain purified α-Syn was similar to that of native recombinant α-Syn (compare **Figures 4A and 8**), confirming that the observed clustering activity is not sensitive to the origin of α-Syn.

## Discussion

The mechanism by which native α-Syn promotes SNARE-complex assembly in vitro and in vivo remains unknown (**Chandra et al., 2005**; **Burré et al., 2010**). Moreover, young αβγ-synuclein triple knockout mice showed no obvious phenotype, but developed neurological impairments during aging and revealed age-dependent impairments in SNARE-complex assembly (**Burré et al., 2010**). Effects on neurotransmission were observed only upon prolonged trains of high-frequency stimulation (**Abeliovich et al., 2000**; **Cabin et al., 2002**) (however, see **Chandra et al., 2004** for different results). Transgenic mice with overexpressed human α-Syn invariably exhibit signs of neurodegeneration, but impairment in synaptic vesicle exocytosis and a reduction in neurotransmitter release were observed only in some studies (**Nemani et al., 2010**; **Scott et al., 2010**) but not in another study (**Burré et al., 2010**).

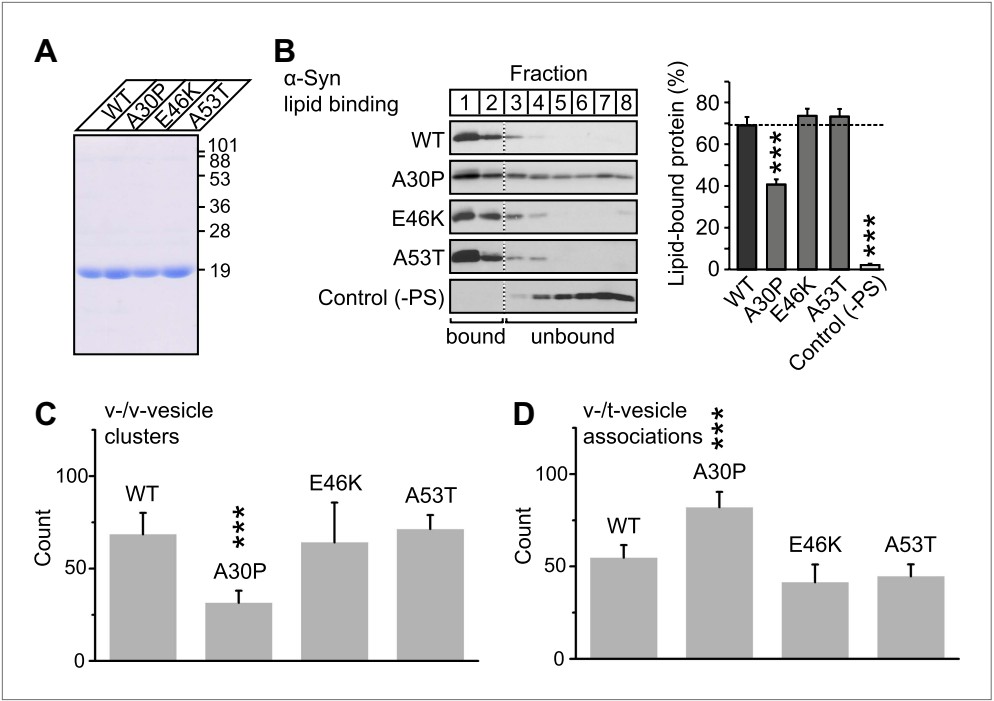

**Figure 7**. V-vesicle clustering correlates with lipid binding of native α-Syn. (**A**) Purification of α-Syn expressed in bacteria. Purified α-Syn wildtype and the three Parkinson's disease (PD)-related mutants A30P, E46K, and A53T (all tag-free constructs) were analyzed by SDS-PAGE and Coomassie staining. (**B**) Lipid binding of recombinant α-Syn wildtype, and of the PD mutants A30P, E46K, and A53T. α-Syn was incubated with negatively charged liposomes (30% phosphatidylserine, 70% phosphatidylcholine, or 100% phosphatidylcholine as control), and subjected to a flotation assay. Eight fractions were collected from top to bottom of the flotation gradient, and equal volumes of each fraction were separated by SDS-PAGE and immunoblotted for α-Syn. Top two fractions were defined as lipid-bound, and quantitated as percent of total α-Syn. Data shown are means ± SEM; n = 3. (**C**) Impaired lipid-binding of α-Syn decreases v-vesicle clustering. In order to measure vesicle clustering, the number of DiI-labeled v-vesicles that were docked to immobilized v-vesicles was counted, as illustrated in **Figure 3**. (**D**) Impaired lipid-binding of α-Syn increases the number of v-/t-vesicle associations. The number of fluorescent spots arising from labeled v-vesicles were counted that were docked to t-vesicles (immobilized to the imaging-surface), as illustrated in **Figure 2**, but in the presence of the specified α-Syn mutants without complexin-1. In all panels, error bars are standard deviations from 10 random imaging locations in the same sample channel, and *** indicates p<0.001 by the Student's t-test.

Overexpression of α-Syn in the substantia nigra pars compacta of rodents using viral vectors also invariably leads to neurodegeneration (**Burré et al., 2012**), and may cause an associated decrease in dopamine release (**Gaugler et al., 2012**; **Lundblad et al., 2012**).

 We tested a possible function of native α-Syn using a reconstituted system consisting of v-vesicles that mimic synaptic vesicles, and t-vesicles that mimic the presynaptic plasma membrane. We found that native α-Syn did not affect the $Ca^{2+}$-triggered fusion efficiency or fusion kinetics on a sub-second timescale (**Figure 1**). The lack of a noticeable effect of native α-Syn on $Ca^{2+}$-triggered fusion in our synthetic system agrees well with extensive in vivo studies with synuclein double and triple knockout mice that showed little effect on synaptic strength (**Chandra et al., 2004**; **Burré et al., 2010**).

 Although there was no effect on $Ca^{2+}$-triggered fusion, we observed clustering of v-vesicles in the presence of native α-Syn (**Figures 2, 4, and 5**). Native α-Syn clustered both synaptobrevin-2 and synaptobrevin-2/synaptotagmin-1 v-vesicles (**Figure 4D**), and the clustering ability was dependent on the synaptobrevin-2 binding domain of α-Syn and the ability of α-Syn to bind to anionic lipids (**Figures 4A,B and 7C**). Furthermore, both recombinant and brain purified native α-Syn induced v-vesicle clustering (**Figures 4 and 8**).

 The observed reduction in the number of associations between v-vesicles (single or multiple) and t-vesicles (**Figure 2A**) was correlated with an increase in the clustering of v-vesicles in the presence of

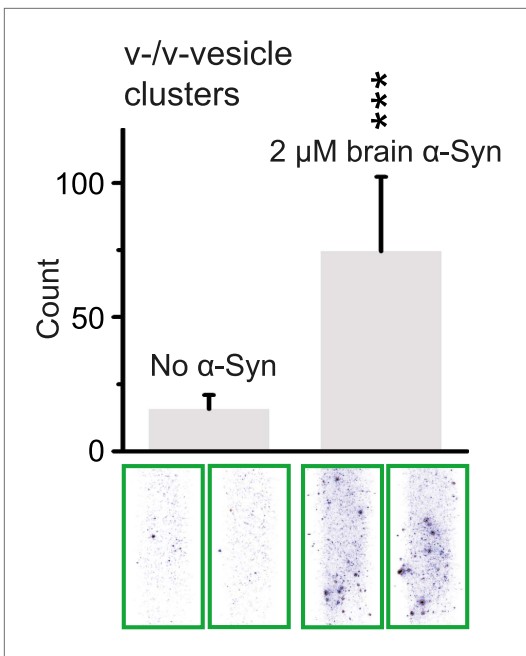

**Figure 8**. Native α-Syn purified from mouse brain induces v-vesicle clustering. Error bars are standard deviations from 20 random imaging locations in the same sample channel. *** indicates p<0.001 by the Student's t-test.

α-Syn (**Figures 4 and 5**). This seemingly paradoxical result can be readily explained since the observed reduction in the number of v-/t-vesicle associations by native α-Syn is due to clustering of v-vesicles into larger particles, which effectively reduces the concentration of individual particles in the sample chamber (**Figure 2C**), and thus reduces the number of fluorescent spots.

The clustering activity of native α-Syn that we uncovered is very different from the effect of dopamine-induced large oligomers/aggregates of α-Syn (**Choi et al., 2013**). We note that although both our v-/t-vesicle assay and the single-vesicle lipid mixing assay by (**Choi et al., 2013**) revealed a reduction of the number of fluorescent ('docked') spots in v-/t-vesicle experiments, the underlying mechanism is different: while native α-Syn clusters v-vesicles and reduces the number of free particles in our v-/t-vesicle assay without interfering with SNARE interactions, large α-Syn oligomers interfere with SNARE complex formation and reduce docking in the lipid mixing assay of (**Choi et al., 2013**).

It has been suggested that clusters of synaptic vesicles could act as a protein buffer (**Denker et al., 2011**) to prevent the loss of accessory proteins involved in vesicle recycling. How could this be accomplished, and also result in an increase of SNARE complex assembly rate as observed by (**Chandra et al., 2005**; **Burré et al., 2010**)? We propose the following model (**Figure 9**): Under normal conditions, native α-Syn would contribute to clustering of synaptic vesicles at the active zone, and thereby assist with the formation of neuronal SNARE complexes by constraining additional synaptic vesicles to be close to the active zone. We found that the Parkinson's disease related A30P point mutant reduced the clustering activity of native α-Syn (**Figure 7C**), suggesting a possible loss-of-function phenotype for this mutant. Under certain other diseased conditions, excess amount of α-Syn could induce severe synaptic vesicle aggregation and thereby result in a reduced readily releasable synaptic vesicle pool, thus affecting neurotransmission.

## Materials and methods

### Native recombinant α-Syn expression and purification

Full-length human α-Syn cDNA was cloned into modified pGEX-KG vectors (GE Healthcare, Uppsala, Sweden), containing an N-terminal TEV protease recognition site. Tag-free and myc-tagged constructs were prepared: The TEV protease cleavage site introduced one extra glycine residue at the N-terminus of the tag-free α-Syn sequence whereas the myc epitope-tagged α-Syn construct encoded the following extra N-terminal residues: GLEEQKLISEEDLGSGS (the myc epitope is underlined). All experiments except those shown in **Figure 4B,D** employed the tag-free α-Syn construct; we used the myc-tag construct for the experiments in **Figure 4B** since the C-terminal truncation mutant of α-Syn[1-95] could not be immuno-detected with our antibody. Mutant α-Syn constructs were generated by site-specific mutagenesis, according to the manufacturer's protocol (Stratagene; Agilent Technologies, Santa Clara, CA). All proteins were expressed as GST fusion proteins in bacteria (BL21 strain), essentially as described (**Burré et al., 2010**). Bacteria were grown to OD 0.6 (measured at 600 nm), and protein expression was then induced with 0.05 mM isopropyl β-D-thiogalactoside (IPTG) for 6 hr at ambient temperature. Bacteria were harvested by centrifugation for 20 min at 4500 g, and pellets were resuspended in solubilization buffer (PBS, 0.5 mg/ml lysozyme, 1 mM PMSF, DNase, and an EDTA-free protease inhibitor cocktail; Roche Diagnostics Corporation, Indianapolis, IN). Cells were

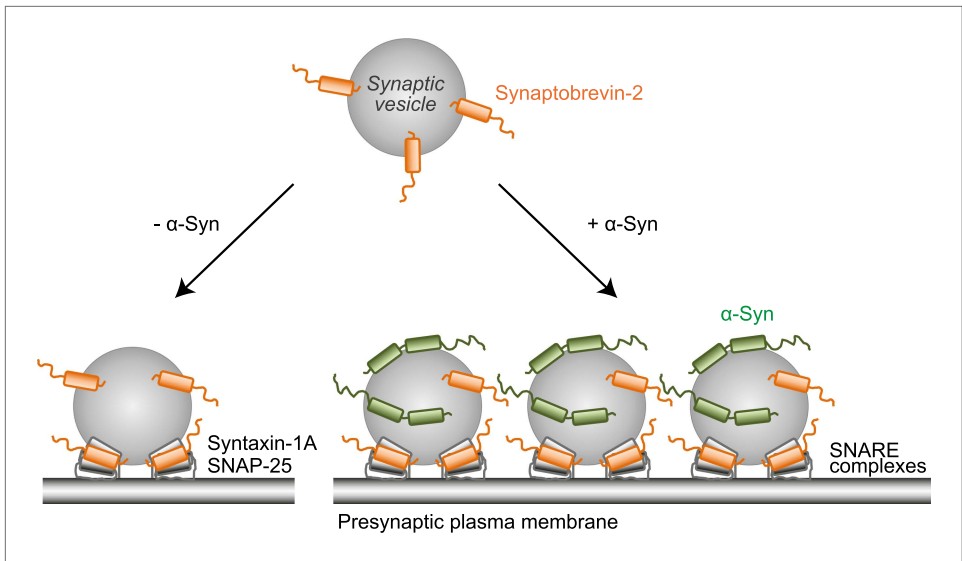

**Figure 9**. Proposed model of native α-Syn function. Native α-Syn binds at the same time to the synaptic vesicle membrane and synaptobrevin-2. Synaptic vesicle fusion with the presynaptic plasma membrane is mediated by the three neuronal SNARE proteins synaptobrevin-2 (on the synaptic vesicle membrane), SNAP-25 and syntaxin-1 (on the presynaptic plasma membrane), which form the synaptic SNARE-complex. Native α-Syn clusters synaptic vesicles depending on binding to synaptobrevin-2 and the vesicles themselves, which may result in a local increase of synaptic vesicles at the presynaptic plasma membrane, and a subsequent increase in SNARE-complex formation.

broken by sonication, and insoluble material was removed by centrifugation for 30 min at 20,000×$g_{av}$ and 4°C. Proteins were affinity-purified using glutathione sepharose bead (GE Healthcare) incubation overnight at 4°C, followed by TEV protease (Invitrogen, Grand Island, NY) cleavage overnight at ambient temperature. His-tagged TEV protease was removed by incubation with Ni-NTA overnight at 4°C. The protein concentration was assessed using BCA method according to the manufacturer's protocol (Thermo Scientific, Rockford, IL). The protein sample was flash-frozen and stored at −80°C.

### Recombinant SNAREs, synaptotagmin-1 and complexin-1 expression and purification

All DNAs encode rat proteins, and were expressed and purified as described by (*Kyoung et al., 2011*) with modifications (*Diao et al., 2012a*). Briefly, his-tagged syntaxin-1A, synaptobrevin-2, SNAP-25A, and synaptotagmin-1 were expressed in *E. coli* and purified using a combination of Ni-NTA affinity (Qiagen, Hilden, Germany) and size exclusion chromatography on a Superdex 200 column (GE Healthcare). Synaptotagmin-1 was further purified using cation exchange chromatography on a Mono-S column (GE Healthcare). His-tags were removed from syntaxin-1A, synaptobrevin-2, and SNAP-25A with TEV protease, or from synaptotagmin-1 with PreScission protease (GE Healthcare). Wildtype complexin-1 was purified essentially as described (*Diao et al., 2012a*) with modifications. Briefly, it was expressed as an N-terminal hexa-his tagged protein from pET28a (Novagen, EMD Chemicals, Gibbstown, NJ) in BL21 (DE3) at 30°C using an auto-induction system (*Studier, 2005*). After binding to Ni-NTA beads, the protein was eluted by overnight cleavage with thrombin. Thrombin was then inactivated with 1 mM PMSF. The cleaved protein was subjected to size exclusion chromatography using a Superdex 200 10/300 column (GE Healthcare) and concentrated to 200 µM. 10% glycerol was added to all purified protein solutions. Complexin-1 and SNAP-25A were flash-frozen and stored as aliquots at −80°C, whereas synaptobrevin-2, synaptotagmin-1, and syntaxin-1A were used from freshly-made preparations.

### Native brain α-Syn purification

Mouse brains homogenized in phosphate-buffered saline (PBS) and protease inhibitors (Roche Diagnostics Corporation, Indianapolis, IN) were fractionated by ultracentrifugation (3 × 280,000×$g_{av}$). α-Syn was purified at 4°C by Q-Sepharose (GE Healthcare) anion exchange chromatography where

α-Syn eluted at 0.3–0.5 M NaCl, 20 mM Tris–HCl pH 7.4, by phenyl-Sepharose (Sigma-Aldrich, St. Louis, MO) hydrophobic interaction chromatography where α-Syn was recovered in the flow-through in 1 M $(NH_4)_2SO_4$, and by gel filtration on a Superdex 200 10/300 GL column (AKTA; GE Healthcare) in PBS. Details can be found in *Burré et al. (2013)*.

## Lipid binding assay

Preparation of vesicles and liposome flotation assay were performed as previously described (*Burré et al., 2010*). Briefly, 1 mg brain phosphatidylcholine (PC; Avanti Polar Lipids, Alabaster, AL) or 0.7 mg PC and 0.3 mg brain phosphatidylserine (PCPS; Avanti Polar Lipids) in chloroform were dried in a glass vial under a nitrogen stream. Residual chloroform was removed by lyophilization for 2 hr. To obtain a solution of 1 mg/ml lipids, 1 ml PBS was added to each vial and vortexed well. Small unilamellar vesicles were formed by sonicating lipid solutions on ice (*Barenholz et al., 1977*). For α-Syn binding to vesicles, 5 μg proteins were incubated with 100 μg vesicles for 2 hr at ambient temperature. Samples were then subjected to a liposome flotation assay as previously described (*Burré et al., 2010*).

## Gel electrophoresis and protein quantitation

Quantitative immunoblotting experiments were performed with iodinated secondary antibodies as described previously (*Rosahl et al., 1995*). Protein samples were separated by SDS-PAGE, and transferred onto nitrocellulose membranes. Blots were blocked in Tris-buffered saline (TBS) containing 0.1% Tween-20 (Sigma-Aldrich) and 3% fat-free milk for 30 min at ambient temperature. The blocked membrane was incubated overnight in blocking buffer containing primary antibody, followed by five washes. The washed membrane was incubated in blocking buffer containing either horse-radish peroxidase (HRP)-conjugated secondary antibody (1:5000; MP Biomedicals, Santa Ana, CA) for 1 hr at ambient temperature, or [125]I-labeled secondary antibodies (1:1000; PerkinElmer, Santa Clara, CA). HRP immunoblots were developed using enhanced chemiluminescence (GE Healthcare). [125]I blots were exposed to a PhosphorImager screen (GE Healthcare) overnight and scanned using a Typhoon scanner (GE Healthcare), followed by quantitation with ImageQuant software (GE Healthcare).

## Primary antibodies

α-Syn (cl. 610786; BD Transduction, San Jose, CA), c-myc (cl. 9E10; Developmental Studies Hybridoma Bank, Iowa City, IA).

## Circular dichroism (CD) spectroscopy

CD spectra were collected with an Aviv model 202-01 spectrometer (Aviv Biomedical, Lakewood, NJ). Spectra are averages of eight replicates collected at 1-nm steps from 195 to 240 nm with a spectral bandwidth of 1 nm and a 1-s averaging time. The spectra of 1.4 μM α-Syn were acquired in pH 7.4, 18 mM NaCl, 4 mM HEPES, 4 μM EGTA, 0.2% β-mercaptoethanol with and without 960 μM lipids in protein-free vesicles that were formed with the same protocol (*Kyoung et al., 2012*) used for the fusion and clustering experiments.

## Single vesicle-vesicle Ca²⁺-triggered fusion experiments

Our experimental setup was similar to that previously described (*Diao et al., 2012a*; *Kyoung et al., 2012*) (*Figure 1A*), except that lipid dye was not used, and the time stamp of Ca²⁺ arrival in the evanescent field was determined by the instance of the first content-mixing event (i.e., stepwise increase of content dye fluorescence intensity) among all docked vesicles.

   T-vesicles (with reconstituted syntaxin-1A and SNAP-25A) and v-vesicles (with reconstituted synaptobrevin-2 and synaptotagmin-1) were prepared as previously described (*Diao et al., 2012a*; *Kyoung et al., 2012*). We showed previously that our particular vesicle preparation and protein reconstitution protocols (*Kyoung et al., 2012*) produce homogeneous populations of vesicles with an average diameter of approximately 80 nm, and physiological protein concentrations (*Kyoung et al., 2011*). The resulting t-vesicle solution was diluted 10× and immobilized on an imaging surface via biotin/neutravidin interactions under saturating conditions. Specifically, 100 μl of the diluted t-vesicle solution was injected into the sample chamber and incubated for 30 min, followed by a 200 μl vesicle buffer exchange (vesicle buffer is defined as 90 mM NaCl, 20 mM HEPES pH 7.4, 20 μM EGTA, 1% β-mercaptoethanol). V-vesicles (with reconstituted synaptobrevin-2 and synaptotagmin-1) containing

self-quenched content dye (sulforhodamine B; Invitrogen) were prepared as previously described (*Diao et al., 2012a*; *Kyoung et al., 2012*). The v-vesicle solution was diluted 150× and added to the immobilized t-vesicle surface with 2 µM complexin-1 in presence or absence of 2 µM α-Syn. The presence of 2 µM α-Syn corresponds to a v-vesicle-lipid to α-Syn-protein ratio of 11:1, with the actual lipid to protein ratio being larger due to the presence of the saturated t-vesicle surface; at this condition, we did not observe destabilization of the v-SNARE vesicles by α-Syn since no leaking of content dyes from docked v-vesicles was observed. The system was incubated for 30 min, followed by buffer exchange (1 × 200 µl vesicle buffer for 6 s), removing unbound or weakly bound complexin and α-Syn. Then, 500 µM $Ca^{2+}$ buffer was injected into the sample chamber, and the fluorescence intensity of the content dye for each docked v-vesicle (but excluding few large fluorescent spots, see arrow in the lower right image of *Figure 2A*) was monitored with total internal reflection (TIR) wide-field microscopy (Nikon Instruments, Melville, NY) using an electron multiplying charge-coupled device (CCD) camera (iXon+ DV 897E; Andor Technology USA, South Windsor, CT). The smCamera program (written in C++) from Taekjip Ha's laboratory was used for data acquisition and analysis as described previously (*Diao et al., 2012a*; *Kyoung et al., 2012*). The number of v-/t-vesicle associations was calculated by counting the number of fluorescent spots prior to $Ca^{2+}$ injection, averaged over the specified number of images in the same sample channel. This simple averaging procedure was possible since the surface coverage was homogenous within the sample channel.

An instance of complete fusion was characterized by a stepwise increase in the content dye fluorescence intensity; this increase resulted from the approximately twofold dilution of the content dye upon fusion between a v- and a t-vesicle, and subsequent dequenching of the dye. For the histograms shown *Figure 1B*, the time stamp of $Ca^{2+}$ injection was defined as the instance of the first content-mixing event among all docked vesicles within a particular field of view. Histograms were combined from several fields of views by using these time stamps for post-synchronization. Histograms were self-normalized with respect to total number of fusion events, and cumulative distributions calculated. Histograms were fitted to bi-exponential decay functions using OriginPro 8.6 (OriginLab, Inc. Northampton, MA).

## Single-vesicle clustering experiments

For the clustering experiments the same v-vesicle preparation and protein reconstitution, and vesicle immobilization protocols were used as for the single vesicle-vesicle fusion experiments described in the previous section, except that lipid dyes were added. Specifically, a saturated layer of 1,1'-dioctadecyl-3,3,3',3'-tetramethylindodicarbocyanine perchlorate, DiIC18(5) (DiD) labeled synaptobrevin-2 containing v-vesicles was immobilized on an imaging surface via biotin/neutravidin interactions. The DiD-labeled synaptobrevin-2 v-vesicle solution (obtained by our reconstitution protocol; *Diao et al., 2012a*; *Kyoung et al., 2012*) was diluted 20×. 100 µl of the diluted vesicle solution was injected into the sample chamber and incubated for 30 min (*Figure 3B*, left panel), followed by buffer exchange (1 × 200 µl vesicle buffer for 6 s). We confirmed that the vesicle-covered surfaces were saturated and produced a homogeneous distribution for each surface preparation with red laser excitation (633 nm) of the DiD-labeled immobilized vesicles (*Figure 3B*, right image). As previously reported, more than 1000 vesicles could be immobilized with this method (*Yoon et al., 2008*).

The immobilized v-vesicle surface was incubated for 30 min at the specified α-Syn concentration (*Figure 3B*, middle panel); note that this is an additional step compared to the v-/t-vesicle fusion experiments described above. This step was followed by buffer exchange (1 × 200 µl vesicle buffer for 6 s), removing unbound or weakly bound α-Syn. Next, 1,1'-dioctadecyl-3,3,3',3'-tetramethylindodicarbocyanine perchlorate, DiIC18(5) (DiI)-labeled synaptobrevin-2 vesicles were also prepared with our reconstitution protocol (*Diao et al., 2012a*; *Kyoung et al., 2012*) and diluted 500×. 100 µl of diluted free-floating DiD-labeled v-vesicle solution was injected into the sample chamber (*Figure 3B*, right panel); note that this v-vesicle solution did not contain α-Syn in order to prevent pre-clustering of this vesicle solution. After an incubation period of 120 min, unbound v-vesicles were removed by buffer exchange (2 × 200 µl vesicle buffer for ~20 s).

Sample slides with multiple channels were monitored in a wide-field TIR fluorescence microscope, data were acquired with a CCD camera, and analyzed with the smCamera program, similar to the single vesicle-vesicle fusion experiments described above. A specified number of images were

taken at random locations within each channel on the quartz slide. Details regarding software, slide assembly, and imaging protocols are described in reference (*Diao et al., 2012b*). The number of v-/v-vesicle interactions (clustering) was determined by counting the number of fluorescent spots from emission of DiI upon excitation at 532 nm (*Figure 3A*). The counts were averaged over the specified number of images at random locations in each sample channel. This simple averaging procedure was possible since the surface coverage was homogeneous. For each set of comparisons between different conditions and/or mutants (*Figures 4A,B,D, 7C,D, and 8*), the same protein preparations and surface preparations (quartz slide with immobilized vesicles) were used, and the conditions were run in separate channels on the same slide; although there was some variation in absolute numbers of counts (e.g., due to the use of tag-free vs tagged a-Syn constructs, *Figure 4A* vs *Figure 4B,D*), the relative differences were statistically similar for different protein preparations.

## Cryo-EM

Frozen-hydrated samples of synaptobrevin-2 v-vesicles in the presence of 20 μM α-Syn were prepared using the procedures for observation in cryo-EM as described previously (*Kyoung et al., 2011*). Briefly, the synaptobrevin-2 v-vesicle solution obtained by our reconstitution protocol (*Diao et al., 2012a*; *Kyoung et al., 2012*) was diluted 5× in the presence of 20 μM α-Syn (resulting in v-vesicle-lipid to α-Syn-protein ratio of 33:1) and then incubated on a glow-discharged lacey Form var/carbon 300 mesh copper grid (Ted Palla, Redding, CA, USA), followed by blotting and plunging into liquid ethane. Thus, the concentration of the vesicles used in the cryo-EM experiments was 100-fold higher than in the single vesicle-vesicle clustering experiments. The frozen-hydrated specimen was subsequently observed at liquid nitrogen temperature using a CM200F electron microscope (FEI, Hillsboro, OR, USA) operating at 200 kV, under low dose conditions. Images were collected at a 50,000× magnification and 1.5 μm under-focus on a 2k × 2k UltraScan 1000 camera (Gatan, Inc., Pleasanton, CA).

## Acknowledgements

We thank Patricia Grob and Eva Nogales for cryo-electron microscopy, Minglei Zhao for stimulating discussions, Mark Padolina and Richard Pfuetzner for performing protein purifications, and Steven Chu for an exciting long-term collaboration that lead to the single-vesicle fusion system.

## Additional information

### Competing interests

ATB, Reviewing editor, *eLife*. The other authors declare that no competing interests exist.

### Funding

| Funder | Grant reference number | Author |
| --- | --- | --- |
| Howard Hughes Medical Institute | | Thomas C Südhof, Axel T Brunger |
| National Institutes of Health | R37-MH63105 | Axel T Brunger |
| National Institutes of Health | NS077906 | Thomas C Südhof |

The funders had no role in study design, data collection and interpretation, or the decision to submit the work for publication.

### Author contributions

JD, JB, Conception and design, Acquisition of data, Analysis and interpretation of data, Drafting or revising the article; SV, Acquisition of data, Analysis and interpretation of data, Drafting or revising the article; DJC, Analysis and interpretation of data, Drafting or revising the article, Contributed unpublished essential data or reagents; MS, Acquisition of data, Analysis and interpretation of data, Contributed unpublished essential data or reagents; MK, Analysis and interpretation of data, Contributed unpublished essential data or reagents; TCS, ATB, Conception and design, Analysis and interpretation of data, Drafting or revising the article

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
