## [Decision Letter]

Thank you for sending your work entitled “α-Synuclein Induces Synaptic Vesicle Clustering Via Binding To Phospholipids and Synaptobrevin-2/VAMP2” for consideration at *eLife*. Your article has been favorably evaluated by a Senior editor and 3 reviewers, one of whom is a member of our Board of Reviewing Editors.

The following individuals responsible for the peer review of your submission want to reveal their identity: Reinhard Jahn (Reviewing editor), and Jose Rizo and Yeon-Kyun Shin (peer reviewers).

The Reviewing editor and the other reviewers discussed their comments before we reached this decision, and the Reviewing editor has assembled the following comments to help you prepare a revised submission.

1) The concentration of synuclein used in the experiments is very high (2–20 μM), resulting in high protein:lipid ratios, raising the possibility that the liposomes are completely covered with protein. We suggest to carry out experiments testing whether such high concentrations are required. A straightforward option would be to run dynamic light scattering experiments on mixed liposome solutions using increasing concentrations of synuclein, but other approaches addressing the same problem are also acceptable.

2) A second related question is whether vesicle aggregation is caused by homooligomerization of synuclein between membranes. While we realize that this question may not be trivial to address, one possible option is to pre-incubate the docked vesicles with synuclein, wash away any unbound material, and then test whether synaptobrevin-liposomes still bind. Although such experiments would not be completely conclusive (particularly if the outcome is negative), this question should be approached by such or other suitable experiments.

3) All referees had difficulties with your model that seems highly speculative and almost counter-intuitive: the data presented in the paper actually show a reduction of docking between v-vesicles and t-vesicles by addition of α-synuclein. If this model is correct, wouldn't one expect a substantial effect of synucleing KO on neurotransmitter release? See the recent paper by Choi et al. (*PNAS*, Feb 19 2013), where the authors argue that vesicle clustering by large α-synuclein oligomers inhibit vesicle fusion and reduce neurotransmitter release.

---

## [Author Response]

*1) The concentration of synuclein used in the experiments is very high (2–20 μM), resulting in high protein:lipid ratios, raising the possibility that the liposomes are completely covered with protein. We suggest to carry out experiments testing whether such high concentrations are required. A straightforward option would be to run dynamic light scattering experiments on mixed liposome solutions using increasing concentrations of synuclein, but other approaches addressing the same problem are also acceptable*.

Unfortunately, dynamic light scattering experiments did not produce statistically meaningful results due to the relatively small effect of small numbers of vesicles clustering in such bulk experiments. However, we would like to point out that the effective concentration in our single vesicle clustering experiments (new Figures 3 and 4) is lower than 2–20 μM. The reason is that α-Syn was added after surface immobilization of acceptor DiD-labeled v-vesicles at 2–20 μM; after the incubation, unbound α-Syn was washed away by buffer exchange. Moreover, it is likely that the buffer exchange also removed some of the v-vesicle-bound α-Syn molecules since kinetic studies by NMR spectroscopy revealed that α-Syn binding to membranes is transient with exchanges occurring in the range of 1–10 s (Bodner et al., 2009). Then, donor DiI-labeled v-vesicles were injected in the absence of free α-Syn. Indeed, as Figure 4A shows, v-vesicle clustering has not reached saturation at 2 μM α-Syn incubation concentration. We added a new illustration (Figure 3B), along with a discussion of this point in the Results section.

*2) A second related question is whether vesicle aggregation is caused by homooligomerization of synuclein between membranes. While we realize that this question may not be trivial to address, one possible option is to pre-incubate the docked vesicles with synuclein, wash away any unbound material, and then test whether synaptobrevin-liposomes still bind. Although such experiments would not be completely conclusive (particularly if the outcome is negative), this question should be approached by such or other suitable experiments*.

We apologize that the description of our protocol was somewhat hidden in the Materials and methods section. We therefore added the above-mentioned new Figure 3B. Indeed, we already did what the reviewers suggested by removing unbound α-Syn prior to adding free v-vesicles to the surface-mmobilized v-vesicles. Furthermore, we have measured α-Syn mediated liposome clustering with liposomes that either contain synaptobrevin-2, or lack synaptobrevin-2 (Figure 4A). Only in the former case did we observe significant v-vesicle clustering, whereas in the absence of synaptobrevin-2, no clustering was observed. Thus, v-vesicle clustering is not caused by homo-oligomerization of α-Syn, but requires specific binding of α-Syn to synaptobrevin-2, as well as binding to anionic lipids.

*3) All referees had difficulties with your model that seems highly speculative and almost counter-intuitive: the data presented in the paper actually show a reduction of docking between v-vesicles and t-vesicles by addition of α-synuclein. If this model is correct, wouldn't one expect a substantial effect of synucleing KO on neurotransmitter release? See the recent paper by Choi et al. (PNAS, 19 Feb 2013), where the authors argue that vesicle clustering by large α-synuclein oligomers inhibit vesicle fusion and reduce neurotransmitter release*.

We apologize that our explanation was unclear. To better explain the observed reduction of fluorescent spots in the v-/t-vesicle fusion experiments, we performed an analysis of the distribution of the fluorescence intensities of the observed fluorescent spots (new Figure 2B). The major peak (<1.5 a.u.) at low florescence intensity corresponds to single donor v-vesicles docked to the imaging surface, while the high intensity tail (>2 a.u.) corresponds to small clusters (doubles, triples, …) of v-vesicles docked to the surface. In the presence of α-Syn, we observed a reduction of the low intensity population and a concomitant increase of higher intensity spots (Figure 2B, insert), along with a few very large spots (arrow in lower right image in Figure 2A). This observation implies the formation of v-vesicle clusters, resulting in a decrease of particle number in the finite volume of our system since the initial concentration of individual v-vesicles is identical for both experiments. A new illustration (Figure 2C) has been added to better illustrate this point. Moreover, we would like to note that extensive studies performed in our and in Pablo Castillo's laboratory failed to observe an effect of the α/β-synuclein double KO and the α/β/γ-synuclein triple KO mice on neurotransmitter release (see Chandra et al., PNAS 2004; Burre et al., Science 2010). Our model does not necessitate a role for α-Syn in regulating release, consistent with the actual physiological results.